# The use of transcranial ultrasound and clinical assessment to diagnose ischaemic stroke due to large vessel occlusion in remote and rural areas

Daria Antipova[1]*, Leila Eadie[1], Stephen Makin[1], Helen Shannon[2], Philip Wilson[1], Ashish Macaden[3]

**1** Centre for Rural Health, University of Aberdeen, Inverness, United Kingdom, **2** Department of Radiology, Raigmore Hospital, NHS Highland, Inverness, United Kingdom, **3** Department of Stroke and Rehabilitation Medicine, Raigmore Hospital, NHS Highland, Inverness, United Kingdom

* daria.antipova@abdn.ac.uk

## Abstract

Rapid endovascular thrombectomy, which can only be delivered in specialist centres, is the most effective treatment for acute ischaemic stroke due to large vessel occlusion (LVO). Pre-hospital selection of these patients is challenging, especially in remote and rural areas due to long transport times and limited access to specialist clinicians and diagnostic facilities. We investigated whether combined transcranial ultrasound and clinical assessment ("TUCA" model) could accurately triage these patients and improve access to thrombectomy. We recruited consecutive patients within 72 hours of suspected stroke, and performed non-contrast transcranial colour-coded ultrasonography within 24 hours of brain computed tomography. We retrospectively collected clinical information, and used hospital discharge diagnosis as the "gold standard". We used binary regression for diagnosis of haemorrhagic stroke, and an ordinal regression model for acute ischaemic stroke with probable LVO, without LVO, transient ischaemic attacks (TIA) and stroke mimics. We calculated sensitivity, specificity, positive and negative predictive values and performed a sensitivity analysis. We recruited 107 patients with suspected stroke from July 2017 to December 2019 at two study sites: 13/107 (12%) with probable LVO, 50/107 (47%) with acute ischaemic stroke without LVO, 18/107 (17%) with haemorrhagic stroke, and 26/107 (24%) with stroke mimics or TIA. The model identified 55% of cases with probable LVO who would have correctly been selected for thrombectomy and 97% of cases who would not have required this treatment (sensitivity 55%, specificity 97%, positive and negative predictive values 75% and 93%, respectively). Diagnostic accuracy of the proposed model was superior to the clinical assessment alone. These data suggest that our model might be a useful tool to identify pre-hospital patients requiring mechanical thrombectomy, however a larger sample is required with the use of CT angiogram as a reference test.

**Data Availability Statement:** All relevant data are within the manuscript and its Supporting Information files.

**Funding:** The authors received no specific funding for this work. Leila Eadie's work was funded by the European Space Agency SatCare grant. The funders had no role in study design, data collection and analysis, decision to publish, or preparation of the manuscript.

**Competing interests:** The authors have declared that no competing interests exist.

# Introduction

The gold standard treatment of acute ischaemic stroke due to large vessel occlusion (LVO) is mechanical thrombectomy which is performed in specialised centres with neurointerventional facilities, ideally within 6 hours of onset [1, 2]. Diagnosis is made with computed tomography angiography (CTA) or magnetic resonance angiography (MRA). There are many remote areas internationally where people do not reach hospital on time to be eligible for reperfusion treatment due to long transport times [3, 4]. In these circumstances, a reliable pre-hospital triage system would allow non-specialist practitioners, such as general practitioners and paramedics, to decide whether to transfer a patient with suspected stroke to a local centre that can only offer therapy with intravenous thrombolysis with tissue plasminogen activator (IV tPA) or to arrange longer and costly travel to a regional centre offering mechanical clot retrieval.

Proposed pre-hospital methods of diagnosis include an ambulance equipped with a computed tomography (CT) scanner and stroke team [5–7] which is not feasible outwith urban areas. Clinical features may indicate the presence of LVO: the presence of cortical signs (gaze deviation, aphasia, and neglect) has 91% sensitivity and 70% specificity when compared to 85% sensitivity and 53% specificity of motor symptoms alone [8]. National Institutes of Health Stroke Scale (NIHSS) score and Rapid Arterial Occlusion Evaluation Scale (RACE) score are the best performing of the currently available tools but their false-negative rates remain higher than 25% [9].

Portable transcranial ultrasound has emerged as a potential method for rapidly assessing the intracranial circulation [10, 11] demonstrating sensitivity ranging from 68% to 100% and specificity 78% to 99% for detecting acute vessel occlusion and stenosis [12]. It can detect haemorrhagic lesions in the deep brain structures [13], and a shift of intracranial midline structures $\geq 0.25$cm can be used as an indicator of a large brain haemorrhage [14]. This is a non-invasive and affordable diagnostic tool that takes as little as 15 minutes for a complete assessment of cerebral vessels [15, 16] and can be performed in space-restricted environments, such as ambulances [17–19].

We hypothesised that a novel triage model combining non-contrast transcranial ultrasound and clinical assessment ("TUCA") could be designed with the aim of selecting patients who would benefit from IV tPA and/or a direct transfer for mechanical thrombectomy. Our primary objective was to compare diagnostic accuracy of the novel triage model with the final discharge diagnosis informed by head CT. For the secondary objective we aimed to determine the proportion of suspected stroke patients in whom transcranial ultrasound has limited diagnostic value due to insufficient acoustic window.

# Materials and methods

We designed an in-hospital exploratory study as part of a larger project with the aim of determining which remote-living patients might benefit from reperfusion therapy (IV tPA and/or mechanical thrombectomy) on the basis of transcranial ultrasound and clinical assessment. Assuming a sensitivity of 90% for detecting haemorrhagic stroke and a two-sided 95% confidence interval (CI) extending 7% on either side of this value (i.e. from 83% to 97%), we calculated that 500 patients would be required to reach an accurate estimate of diagnostic accuracy for excluding intracranial haemorrhage [20] as an absolute contraindication for reperfusion therapy. We were guided by the STARD statement for reporting the results of diagnostic accuracy studies [21].

## Subjects

We consecutively recruited patients with symptoms of acute stroke who presented to a district general hospital (Raigmore Hospital, Inverness, UK) between July 2017 and December 2019, and the Queen Elizabeth University Hospital in Glasgow, UK from October to December 2019. We included patients who presented with symptoms suggestive of sudden onset of acute stroke within the last 72 hours and who had urgent CT brain requested. We also included patients with suspected subarachnoid haemorrhage (SAH) as this condition might present to rural clinicians with symptoms suggestive of a stroke. Consent was gained from a relative or legal representative in compliance with the Adults with Incapacity Act (Scotland, 2000). We excluded patients under the age of 16 and those who did not have urgent CT brain imaging requested.

## Data collection

Baseline demographic and clinical data were collected retrospectively from clinical notes. The NIHSS total score was calculated using standard procedure [22]. We also collected data on whether the patient had already been treated with IV tPA (as thrombectomy was not yet available in Scotland).

Final discharge diagnosis informed by CT was used to classify patients as: (A) acute stroke probably due to LVO; (B) acute ischaemic stroke with no evidence of LVO; (C) haemorrhagic stroke; (D) stroke mimicking conditions and transient ischaemic attacks (TIAs). A TIA was defined as a focal neurological deficit when all symptoms resolve within 24 hours with no evidence of ischaemia on CT. The final diagnosis was made at the patient's hospital by the treating clinician who may have been a general physician or stroke specialist and was used as the "gold standard".

## Brain imaging

CT—the current recommended brain imaging in UK stroke guidelines [2]—was the reference imaging test. It was performed as per standard hospital protocol upon patient presentation using a General Electric 750HD G64-slice Discovery or General Electric 64-slice light speed scanner. CTA was not performed as part of routine clinical care in the district hospital.

Non-contrast transcranial colour-coded sonography (TCCS) was performed within 24 hours of the CT scan using a SonoSite M-Turbo® Point-of-Care ultrasound machine; Philips Sparq or Philips CX50 ultrasound. A 1–5 MHz, low-frequency phased array transducer with a small footprint was used to facilitate ultrasound penetration through the skull. All TCCS scans in each recruitment site were recorded by the same research team members—either a neurologist with more than two years of experience in transcranial ultrasonography or a sonographer with over 20 years of experience. The researcher performing TCCS was blinded to the CT findings but not clinical features. Results of the index test (TCCS) were not available to the assessors of the reference standard (CT) although clinical information was provided. Transcranial ultrasonography results were interpreted by a neurologist who performed TCCS; in complex cases further advice was sought from a radiologist and a rehabilitation medicine and stroke physician.

Patients underwent TCCS whilst in the supine or sitting position. Scanning was done at the patient's bedside through both temporal bone windows using TCCS to visualise brain structures and to assess blood flow in the major intracranial arteries. The operator followed a standard protocol to explore for:

1. Brain structures in the coronal and transverse planes, aiming to visualise any acute intracerebral haemorrhage as a homogenous hyperechogenic area well distinguished from surrounding tissues [13]; spontaneous SAH might be occasionally detected as hyperechogenicity in the basal cisterns [23].

2. Midline shift, by taking three consecutive measurements of the distance between the ultrasound probe and the centre of the third ventricle measured along a line perpendicular to the walls of the third ventricle from the ipsilateral side. Midline shift was calculated using the equation reported by Stolz et al. [24] as the difference in distance from the temporal bone to the middle of the third ventricle measured on the ipsilateral and contralateral sides and divided by 2.

3. Visualisation of the cerebral peduncles / brainstem to locate the circle of Willis.

4. Visualisation of the blood flow in the major intracranial arteries forming the circle of Willis bilaterally with colour-coded sonography—middle cerebral artery (MCA), anterior cerebral artery (ACA) and posterior cerebral artery (PCA). MCA was identified as "orthograde" blood flow (towards the ultrasound probe) at insonation depth of 40-65mm and 30-40mm for proximal (M1) and distal (M2) segment, respectively. ACA was detected as "retrograde" flow (away from the ultrasound probe) at the 60-75mm depth of insonation. PCA was visualised as "orthograde" (P1 segment) or "retrograde" (P2 segment) at 55-75mm depth [15].

LVO was defined as occlusion of one of the major intracranial arteries forming the circle of Willis, namely the M1 or M2 segments of the MCA, ACA and PCA, when the flow was absent or minimal, blunted, or damped throughout these vessels [25–31].

An insufficient temporal bone acoustic window was defined as the inability to visualise both the heart-shaped cerebral peduncles as a hypoechoic structure and the contralateral temporal skull bone.

## Statistical analysis

A statistical analysis plan was developed and published on OSF, a free, open online platform, prior to commencing the final analysis and is available at: https://osf.io/dkwy8/. The analysis was carried out using IBM SPSS Statistics 25.

We constructed two regression models based on transcranial ultrasound and clinical assessment variables: a binary model for diagnosis of haemorrhagic stroke and an ordinal model for diagnosis of acute stroke due to LVO, acute stroke with no evidence of LVO, and non-stroke (in the order reflecting treatment options from the most invasive, i.e. mechanical thrombectomy, to the least—no intervention). Transcranial ultrasound and clinical candidate predictors have been chosen as those considered clinically significant based on previously published data and specialist experience. In order to avoid overfitting of the regression models, we performed a correlation analysis using Spearman correlation for skewed data to exclude independent variables that were highly correlated with each other, i.e. when coefficient reached 0.4 and p-value was $\leq 0.05$.

Receiver operating characteristic (ROC) curves were constructed from the models and discharge diagnosis using SPSS with computed probabilities of combined variables which were used as a test variable to be displayed as a single ROC curve. Area under the curve (AUC) with 95% CI was calculated using SPSS. True positive (TP), true negative (TN), false positive (FP), and false negative (FN) cases were identified using optimal cut-off points as 2x2 classification tables. Sensitivity, specificity, positive (PPV) and negative predictive values (NPV) were calculated.

We compared the accuracy of the model based on transcranial ultrasound plus clinical assessment to the one based on clinical findings alone. Sensitivity analysis was carried out to test the robustness of the results given some diagnostic uncertainty due to insufficient acoustic window and to improve understanding of the role of the time interval between symptom onset and transcranial ultrasound.

### Designing the tentative triage model

A tentative triage model was designed on the basis of clinically significant transcranial ultrasound and clinical variables, and evidence of any contraindications for IV tPA [32]. The aim of the proposed model was to select patients with: (1) probable acute ischaemic stroke due to LVO that could benefit from mechanical thrombectomy with or without IV tPA; (2) probable acute ischaemic stroke with or without signs of LVO that could benefit from IV tPA only; (3) probable intracranial haemorrhage which would be considered as an absolute contraindication for reperfusion therapy.

### Ethical approval and consent to participate

This study was granted ethical approval from Scotland A REC 17/SS/0046, and approved by NHS Highland Research Management (reference 1225), NHS Greater Glasgow & Clyde Board (reference GN19ST364) and NHS Grampian Research & Development management (reference 2019ST006; data collection in this site has not started). Consent for publication was obtained from participants or their relative / legal representative according to the Adults with Incapacity Act (Scotland, 2000). We obtained written informed consent; in cases where the patient's representative was not physically present within 8 hours of the patient's arrival, verbal consent via telephone was obtained and then followed with written consent at a later timepoint.

## Results

### Characteristics of participants

We recruited 107 patients suspected of having suffered an acute stroke from July 2017 to December 2019. A patient flow summary is presented in Fig 1. No patients withdrew from the study.

We recruited 63 patients with acute ischaemic stroke: 50 subjects with acute stroke without LVO and 13 patients with probable LVO, including occlusion in the M1 (n = 5) (Fig 2), in the M2 (n = 5), in the distal MCA (n = 1), and in the PCA (n = 2).

Intracerebral haemorrhage was diagnosed in 18 patients (Fig 3); one of the recruited participants had a final diagnosis of SAH.

Final diagnosis of a TIA was made in 13 cases; 13 patients were diagnosed with a stroke mimicking condition, such as multiple sclerosis, benign paroxysmal positional vertigo, functional neurological disorder, neuropathic pain, Bell's palsy, migraine, hypertension induced paraesthesia, and angioma. Insufficient acoustic window was identified in 18/107 (17%) patients.

Patients' baseline characteristics are presented in Table 1.

All patients were able to tolerate transcranial ultrasound examination; time to complete the transcranial ultrasound scan varied from 7 to 49 minutes (median 20 minutes, IQR = 16–27 minutes) mainly depending on the acoustic window availability. Time from symptom onset to transcranial ultrasound varied from 2 hours 28 minutes to 72 hours (median 24 hours, IQR = 19 hours 3 minutes to 28 hours 30 minutes). All patients except one underwent

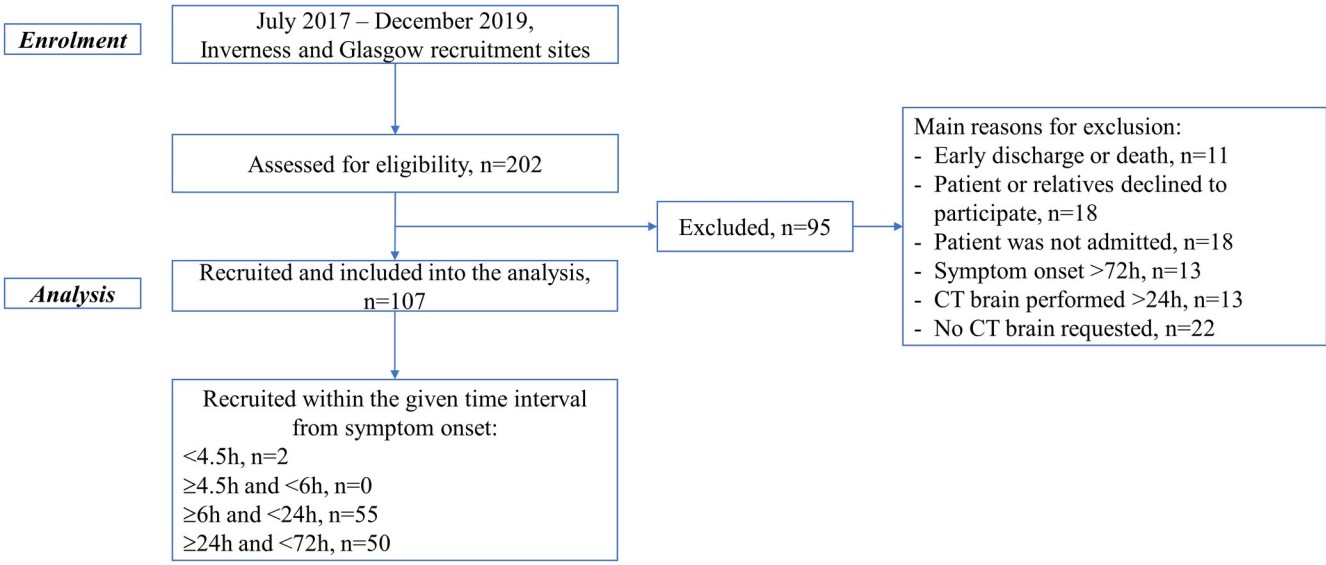

**Fig 1. Flow-diagram.** The recruitment of participants into the study and final analysis.

transcranial ultrasound scanning after the CT. Time interval between CT brain scan and transcranial ultrasound varied from 58 minutes to 24 hours (median 16 hours 30 minutes, IQR = from 8 hours 3 minutes to 20 hours 27 minutes). Diagnostic accuracy of transcranial ultrasonography in suspected stroke patients is presented in Fig 4.

Transcranial ultrasound detected signs of LVO in 54% (7/13) of ischaemic stroke with probable LVO cases, and missed approximately 31% (4/13) cases (the remaining 15% (2/13) of patients had insufficient temporal acoustic window). Intracranial haemorrhage was seen on ultrasound scans in 56% (10/18) of patients with haemorrhagic stroke; it failed to demonstrate any evidence of brain bleed in 33% (6/18). TCCS provided false positive signs of LVO in 10% (5/50) of patents with ischaemic stroke without LVO and 12% (3/26) of non-stroke patients. False positive signs of intracranial haemorrhage were seen in 2% (1/50) of patients with acute ischaemic stroke.

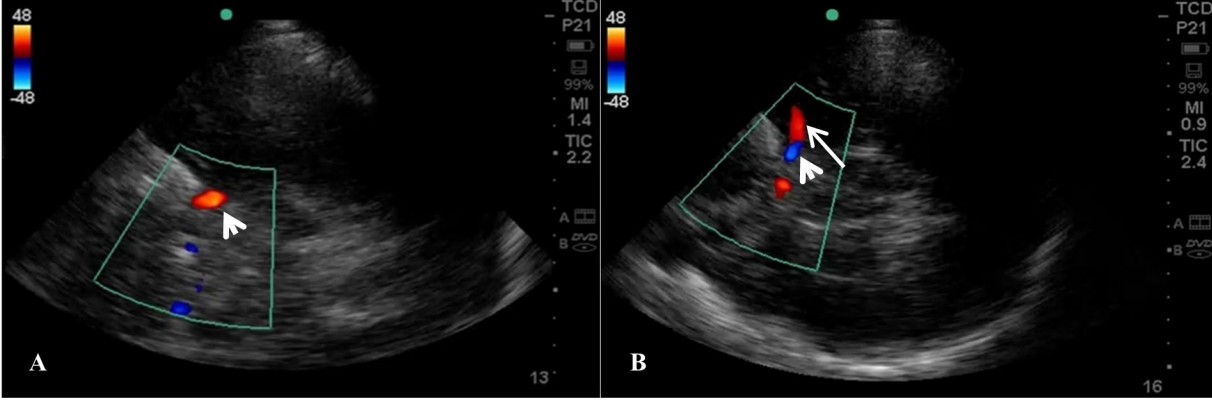

**Fig 2. Transcranial colour-coded sonography findings in a patient with a massive right middle cerebral artery territory infarct.** Absent flow in the middle cerebral artery and reversed flow in the ipsilateral anterior cerebral artery (A) when compared to the flow within the vessels on the unaffected side (B). Long white arrow indicates blood flow within the middle cerebral artery and short white arrow indicates blood flow within the anterior cerebral artery.

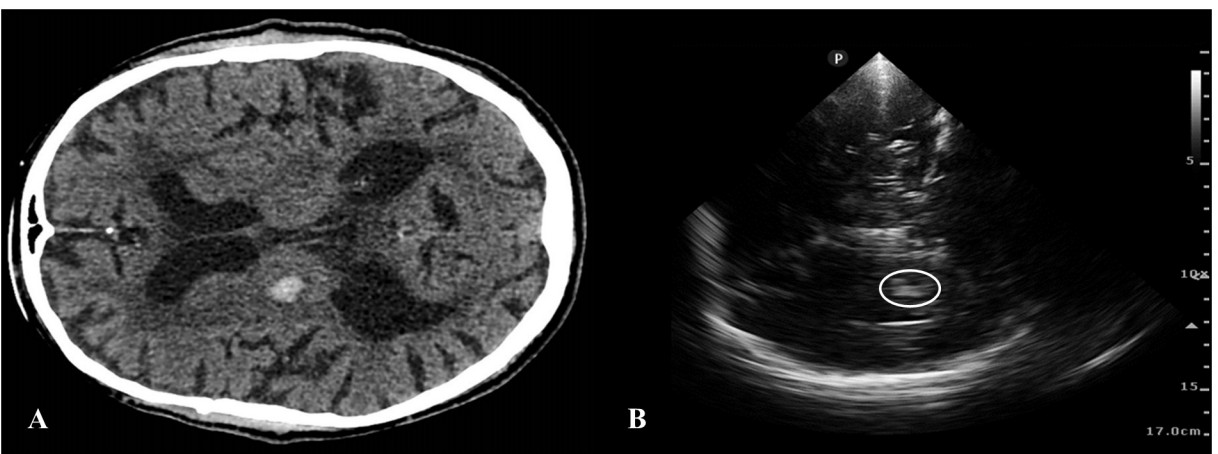

**Fig 3. Brain imaging findings in a patient with acute intracerebral haemorrhage.** CT brain scan (A) and transcranial ultrasound (B) showing a 12 mm haemorrhagic lesion in the right basal ganglia.

Intracranial CTA was performed for five patients with intracerebral haemorrhage, one patient with SAH looking for the source of the bleed, four subjects with acute ischaemic stroke with no evidence of LVO, and three patients with a TIA. Eleven patients received IV tPA prior to the transcranial ultrasound scanning (time lag 6 hours 25 minutes to 24 hours 16 minutes).

**Table 1. Comparison of baseline characteristics between the patient groups.**

|  | Ischaemic stroke due to LVO (n = 13) | Ischaemic stroke without LVO (n = 50) | Haemorrhagic stroke (n = 18) | Stroke mimics and TIA (n = 26) | Total |
|---|---|---|---|---|---|
| **1. Demographic characteristics:** | | | | | |
| 1.1. Male (n, %) | 9 (69%) | 28 (56%) | 12 (67%) | 13 (50%) | 62 (58%) |
| 1.2. Age, median years (IQR) | 66 (61–81) | 73 (68–78) | 76 (71–84) | 65 (50–75) | 72 (64–78) |
| **2. Transcranial ultrasound data:** | | | | | |
| 2.1. Presence of haemorrhage (n, %) | 0 | 1 (2%) | 10 (63%) | 0 | 11 (12%) |
| 2.2. Midline shift median, cm (IQR) | 0.13 (0.08–0.29) | 0.11 (0.04–0.2) | 0.2 (0.06–0.28) | 0.1 (0.03–0.16) | 0.11 (0.05–0.21) |
| 2.3. Signs of LVO (n, %) | 7 (64%) | 5 (12%) | 0 | 3 (13%) | 15 (17%) |
| 2.4. Insufficient window (n, %) | 2 (15%) | 9 (18%) | 4 (22%) | 3 (12%) | 18 (17%) |
| **3. Clinical data:** | | | | | |
| 3.1. Current anti-platelet or anti-coagulant therapy (n, %) | 4 (31%) | 19 (38%) | 6 (33%) | 6 (23%) | 35 (33%) |
| 3.1.1. Anti-platelet therapy (n, %) | 2 (15%) | 10 (20%) | 1 (6%) | 3 (12%) | 16 (15%) |
| 3.1.2. Anti-coagulant therapy (n, %) | 2 (15%) | 9 (18%) | 5 (28%) | 3 (12%) | 19 (18%) |
| 3.2. Presence of cortical signs—gaze deviation, neglect, dysphasia or aphasia, homonymous hemianopia (n, %) | 11 (85%) | 26 (52%) | 9 (50%) | 2 (8%) | 48 (45%) |
| 3.3. NIHSS score, median (IQR) | 8 (3–17) | 4 (2–6) | 7 (2–16) | 1 (0–2) | 3 (1–7) |
| 3.4. mRS score, median (IQR) | 2 (1–5) | 2 (1–3) | 4 (0–5) | 1 (0–1) | 1 (1–3) |
| 3.5. History or ECG findings of atrial fibrillation (n, %) | 5 (39%) | 12 (24%) | 5 (28%) | 3 (12%) | 25 (23%) |
| 3.6. Systolic blood pressure (mmHg), median (IQR) | 145 (126–182) | 152 (135–174) | 168 (150–189) | 150 (136–176) | 152 (137–176) |

Abbreviations: ECG—electrocardiogram; IQR—interquartile range; LVO—large vessel occlusion; mRS—modified Rankin scale; NIHSS—National Institutes of Health Stroke Scale; TIA—transient ischaemic attack.

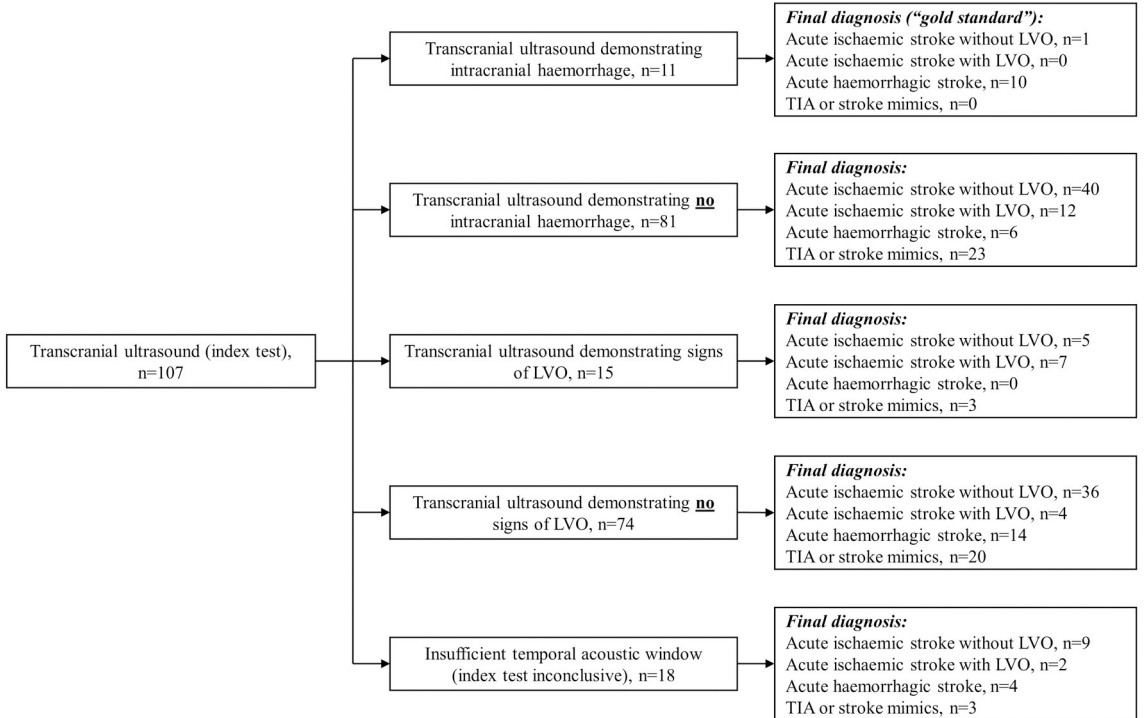

**Fig 4. Diagnostic accuracy of transcranial ultrasonography in suspected acute stroke patients (modified STARD 2015 flow diagram).** Abbreviations: LVO—large vessel occlusion; TIA—transient ischaemic attack.

## Results of the statistical analysis

We chose eight clinically significant variables for the regression models among transcranial ultrasound and clinical candidate variables: presence of haemorrhage on transcranial ultrasound, midline shift, signs of LVO on TCCS, active anti-platelet or anti-coagulant therapy, positive cortical signs, history or ECG findings of atrial fibrillation, age, and systolic blood pressure. The following candidate variables showed significant multicollinearity: NIHSS and mRS (Spearman coefficient 0.7, p<0.01); NIHSS and cortical signs (Spearman coefficient 0.6, p<0.001); cortical signs and mRS (Spearman coefficient 0.4, p<0.001), active use of anti-platelet or anticoagulant therapy and age (Spearman coefficient 0.4, p<0.001).

**Binary logistic regression model for diagnosis of intracranial haemorrhage.** We used a binary logistic regression model to identify odds ratios and p-values of transcranial ultrasound and clinical candidate variables for diagnosis of intracranial haemorrhage. Only 82% (88/107) of cases were included in the analysis (patients with haemorrhagic stroke, n = 16) due to missing transcranial ultrasound data in cases with insufficient acoustic window—given the low number of haemorrhagic cases and the high percentage of missing data, we decided not to construct a binary logistic regression model for diagnosis of acute haemorrhage stroke.

Five clinically relevant variables were used to assess association with intracranial haemorrhage: presence of haemorrhage on transcranial ultrasound, midline shift, active anti-platelet or anticoagulant therapy, cortical signs and systolic blood pressure. The ROC curves demonstrate diagnostic performance of the model based on combined transcranial ultrasound and clinical assessment compared to clinical findings alone (Fig 5).

Fig 5A illustrates two possible cut-off points: one has around 0.81 sensitivity and 0.92 for specificity, and the other one has around 0.84 sensitivity and 0.79 for specificity; the optimal

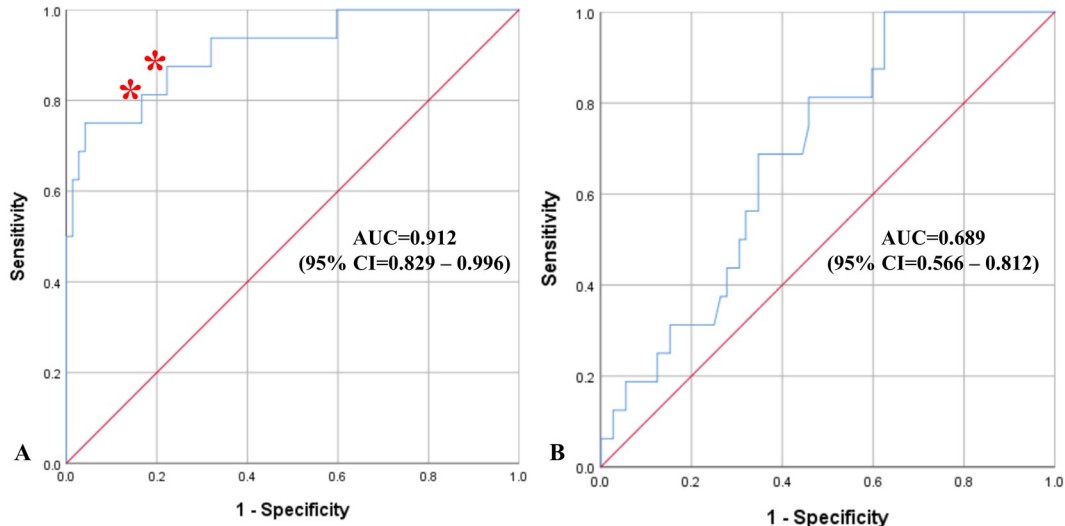

**Fig 5. Receiver operating characteristics plots with diagonal reference line (red) demonstrating diagnostic accuracy of the regression models for diagnosis of acute haemorrhagic stroke.** (A) based on combined transcranial ultrasound and clinical assessment, versus (B) based on clinical findings alone. Red asterisks indicate two possible cut-off points. Abbreviations: AUC—area under the curve; CI—confidence interval.

cut-off may be chosen depending on the preferred sensitivity and specificity values, although the difference between these two cut-off points is minimal.

Diagnostic accuracy of the proposed system and its comparison with the model based on clinical assessment alone is presented in Table 2.

Transcranial ultrasound data improved the percentage of correctly identified haemorrhagic stroke cases by 10% (sensitivity increased by 57%, PPV–by 41%, and NPV increased by 9%; specificity remained unchanged). The false negative (missed intracranial haemorrhage) rates

**Table 2. Diagnostic accuracy of the prognostic model for diagnosis of intracranial haemorrhage.**

**1. Model based on transcranial ultrasound and clinical assessment (cases included in the analysis, n = 88)**

| Observed diagnosis | | Predicted diagnosis | | |
|---|---|---|---|---|
| | | Ischaemic stroke, TIA or stroke mimics | ICH | Percentage correct |
| | Ischaemic stroke, TIA or stroke mimics | 71 | 1 | 99 |
| | ICH | 6 | 10 | 63 |

**Overall percentage correct = 92%**
Sensitivity = 63%; specificity = 99%; PPV = 91%; NPV = 92%

**2. Model based on clinical assessment alone (cases included in the analysis, n = 88)**

| Observed diagnosis | | Predicted diagnosis | | |
|---|---|---|---|---|
| | | Ischaemic stroke, TIA or stroke mimics | ICH | Percentage correct |
| | Ischaemic stroke, TIA or stroke mimics | 71 | 1 | 99 |
| | ICH | 15 | 1 | 6 |

**Overall percentage correct = 82%**
Sensitivity = 6%; specificity = 99%; PPV = 50%; NPV = 83%

Predicted probability cut-off value is 50%. Abbreviations: ICH—intracranial haemorrhage; NPV—negative predictive value; PPV—positive predictive value; TIA—transient ischaemic attack.

**Table 3. Ordinal logistic regression model for diagnosis of acute ischaemic stroke with probable LVO, acute stroke with no evidence of LVO, and non-strokes based on transcranial ultrasound and clinical variables.**

| | Estimate | Standard error | Wald | P-value | Odds ratio | 95% CI for odds ratio | |
|---|---|---|---|---|---|---|---|
| | | | | | | Lower | Upper |
| *Transcranial ultrasound variable* | | | | | | | |
| Presence of LVO signs | 1.663 | 0.659 | 6.368 | 0.012 | 5.278 | 1.450 | 19.211 |
| *Clinical variables* | | | | | | | |
| **Cortical signs** | 2.902 | 0.687 | 17.823 | **0.00002** | 18.207 | 4.733 | 70.037 |
| History or ECG findings of atrial fibrillation | 0.198 | 0.65 | 0.093 | 0.761 | 1.219 | 0.341 | 4.354 |
| Age, years | 0.042 | 0.021 | 3.964 | 0.046 | 1.043 | 1.001 | 1.086 |

Pearson goodness-of-fit test: Chi-square = 118.492; df = 126; p-value = 0.67.

Deviance goodness-of-fit test: Chi-square = 101.426; df = 126; p-value = 0.947.

Pseudo $R^2$: Cox and Snell 0.407; Nagelkerke 0.474.

All variables were included in the model because they all were considered clinically significant. Abbreviations: CI—confidence interval; df—degrees of freedom; ECG—electrocardiogram; LVO—large vessel occlusion.

of the proposed model were lower compared to the clinical assessment alone– 7% and 17%, respectively; there was no difference in false positive rates between the two models (1%).

**Ordinal logistic regression model for diagnosis of acute ischaemic stroke with probable LVO, acute ischaemic stroke with no evidence of LVO, and non-strokes.** An ordinal logistic regression model was constructed for diagnosis of acute stroke with probable LVO, stroke with no evidence of LVO, and non-strokes; patients with haemorrhagic stroke were excluded from this model because in clinical practice these patients would not be considered for reperfusion therapy as intracranial bleed is one of the absolute contraindications. Transcranial and clinical variables, such as presence of haemorrhage on transcranial ultrasound, midline shift, systolic blood pressure values and current anticoagulants or anti-platelet therapy were excluded from the model because they represent prognostic factors for haemorrhagic stroke, which was removed from this analysis. We excluded two other variables due to their significant correlation with cortical signs—mRS and NIHSS scores. Four candidate variables—signs of LVO on transcranial ultrasound, ECG signs or history of atrial fibrillation, age, and cortical signs—were included into the model; their odds ratios and p-values are presented in Table 3.

Seventy five cases were included in the analysis (75/89, 84%)–acute ischaemic stroke due to LVO, n = 11; acute ischaemic stroke without LVO, n = 41; TIA and stroke mimics, n = 23; 14 cases with insufficient acoustic window (14/89, 16%) were not included in the analysis due to missing transcranial ultrasound data.

We used binary logistic regression to calculate diagnostic accuracy of the model for detection of acute ischaemic stroke with probable LVO based on combined transcranial ultrasound and clinical assessment and to compare the accuracy of the proposed model to the one based on clinical findings alone. The ROC curves illustrate diagnostic accuracy of these models with plotted sensitivity and 1-specificity comparing difference between performances of the models with combined transcranial ultrasound and clinical variables versus clinical data only (Fig 6).

Inclusion of transcranial ultrasound variables noticeably improved sensitivity, PPV and NPV of the regression model in comparison with the conventional one based on clinical data alone, although specificity slightly decreased by 3%. Diagnostic accuracy of the regression models for acute ischaemic stroke with signs of LVO is presented in 2x2 tables below (Table 4).

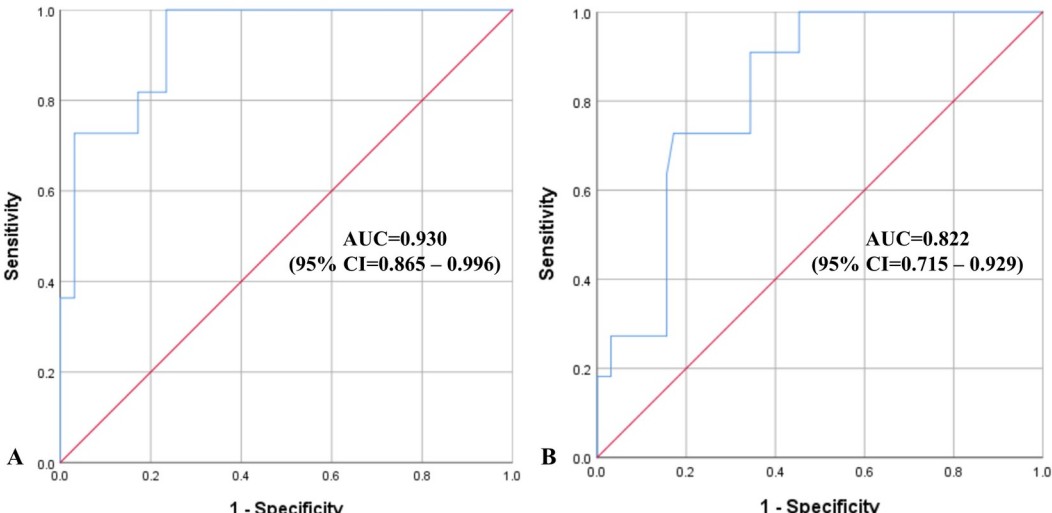

**Fig 6. Receiver operating characteristics plots with diagonal reference line (red) demonstrating diagnostic accuracy of the regression models for diagnosis of acute ischaemic stroke due to LVO.** (A) based on combined transcranial ultrasound and clinical assessment, versus (B) based on clinical findings alone. Abbreviations: AUC—area under the curve; CI—confidence interval; LVO—large vessel occlusion.

The false negative (missed LVO cases) rates of the model based on transcranial ultrasound and clinical assessment were lower compared to the clinical assessment alone– 7% and 15%, respectively; the model showed 3% false positive rates.

**Sensitivity analysis.** We tested the robustness of the results of the regression models with several sensitivity analyses (S1 Table). This was performed to calculate the outcome—likely

**Table 4. Diagnostic accuracy of the prognostic model for diagnosis of acute ischaemic stroke due to LVO based on combined transcranial ultrasound and clinical assessment versus clinical findings alone.**

**1. Model based on transcranial and clinical assessment (cases included in the analysis, n = 75)**

| Observed diagnosis | Predicted diagnosis | | |
|---|---|---|---|
| | Acute ischaemic stroke due to LVO | Acute ischaemic stroke with no signs of LVO; stroke mimics and TIAs | Percentage correct |
| Acute ischaemic stroke due to LVO | 6 | 5 | 55 |
| Acute ischaemic stroke with no signs of LVO; stroke mimics and TIAs | 2 | 62 | 97 |

**Overall percentage correct = 91%**
Sensitivity = 55%; specificity = 97%; PPV = 75%; NPV = 93%

**2. Model based on clinical assessment alone(cases included in the analysis, n = 75)**

| Observed diagnosis | Predicted diagnosis | | |
|---|---|---|---|
| | Acute ischaemic stroke due to LVO | Acute ischaemic with no signs of LVO; stroke mimics and TIAs | Percentage correct |
| Acute ischaemic stroke due to LVO | 0 | 11 | 0 |
| Acute ischaemic stroke with no signs of LVO; stroke mimics and TIAs | 0 | 64 | 100 |

**Overall percentage correct = 85%**
Sensitivity = 0%; specificity = 100%; PPV = 0%; NPV = 85%

Predicted probability cut-off value is 50%. Abbreviations: LVO–large vessel occlusion; NPV–negative predictive value; PPV—positive predictive value; TIA—transient ischaemic attack.

diagnosis based on transcranial ultrasound and clinical assessment—depending on (1) the time interval of under 24 hours from symptom onset to the transcranial ultrasound; (2) including or excluding patients with insufficient temporal acoustic window using theoretical results of the transcranial ultrasound—present or absent signs of intracranial haemorrhage, and present or absent signs of LVO; and (3) excluding patients who were given IV tPA prior to the transcranial ultrasound.

The regression model for diagnosis of intracranial haemorrhage in the first 24 hours from symptom onset showed improved specificity, PPV and NPV but decreased sensitivity (by 6%).

Performance of the regression model for detecting acute ischaemic stroke due to LVO within the first 24 hours from the onset of symptoms also improved. However, using theoretical results of transcranial ultrasound showing present or absent signs of LVO resulted in an overall decreased accuracy of the model by 1%: NPV, sensitivity and specificity decreased by 1%, and PPV decreased by 5%.

## Designing a tentative "TUCA" model for pre-hospital triage of suspected stroke patients based on transcranial ultrasound and clinical assessment for remote and rural communities

We designed a tentative triage model on the basis of relevant transcranial ultrasound and clinical variables ("TUCA"), and evidence of any contraindications for IV tPA [32], where treatment decision could be guided by clinical assessment alone in case of inadequate temporal acoustic window (Fig 7).

## Discussion

We found that the model based on transcranial and clinical assessment was able to identify 55% of patients with probable LVO who would benefit from mechanical thrombectomy and 97% of cases who would not require a direct transfer for the endovascular treatment. This model would have allowed us to reliably exclude 63% of patients with haemorrhagic stroke (sensitivity 63%, specificity 99%, PPV 91%, NPV 92%). The rate of missed haemorrhagic stroke cases improved by 10% with added transcranial ultrasound assessment indicating that approximately 7% (6/88) of haemorrhagic stroke cases might not be identified correctly with the proposed model and may be at risk of getting IV tPA in the absence of definitive imaging, or being inappropriately transferred to an endovascular centre. The number of patients without intracranial haemorrhage that were incorrectly identified as having suffered haemorrhagic stroke (false positive) remained at 1% regardless of the model employed, leading to potential missed opportunities for prompt transfer for reperfusion therapy. Results of the sensitivity analysis demonstrated that diagnostic accuracy of the regression model improved when used within the first 24 hours of symptom onset and in the absence of acoustic window failure.

Performance of the model for diagnosis of acute ischaemic stroke due to LVO improved with the addition of transcranial ultrasound findings in comparison with clinical assessment alone, with an 8% reduction in the false negative rate. Sensitivity analysis of the proposed model's performance in the first 24 hours from symptom onset demonstrated improved sensitivity, specificity and PPV values, with increases of 4%, 3% and 25%, respectively. We excluded patients with confirmed intracranial haemorrhage from the ordinal regression model because it is not sufficiently robust at this stage but we would aim to construct a single model for pre-hospital triage of all suspected stroke patients based on data collected from a larger sample.

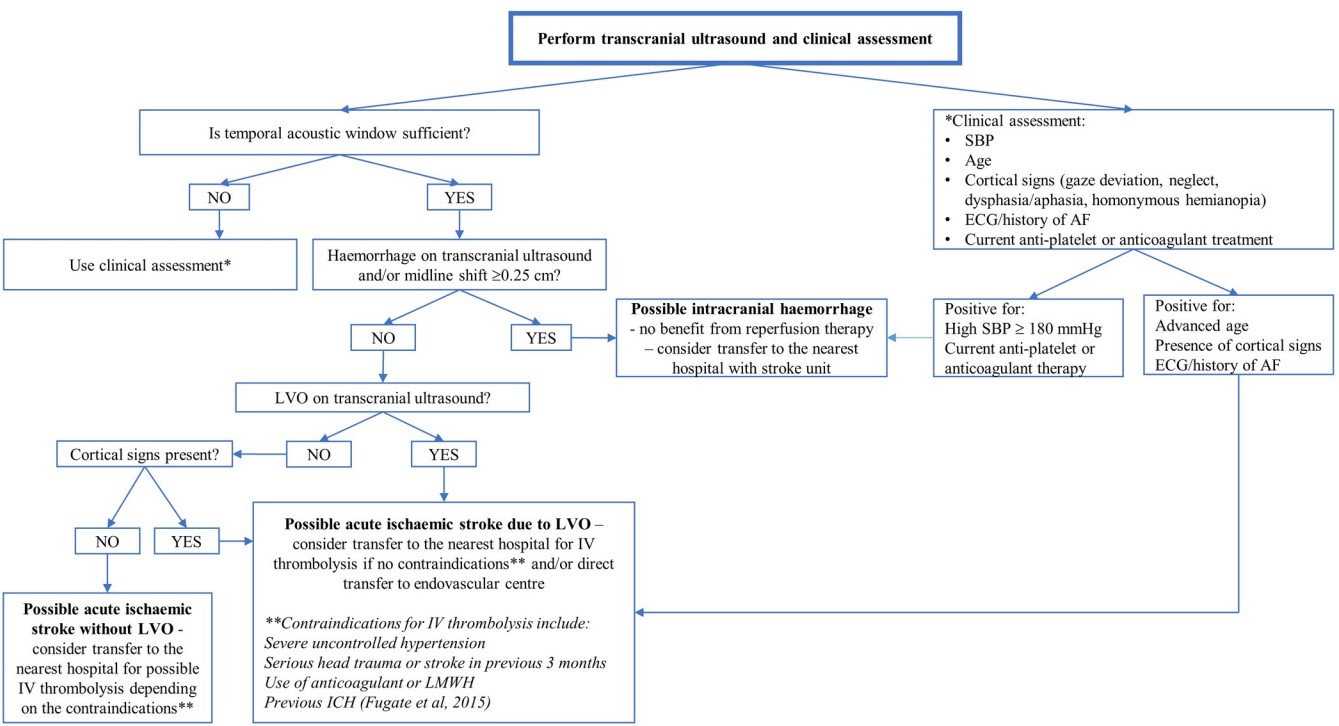

**Fig 7. Tentative "TUCA" model for pre-hospital triage of suspected stroke patients based on transcranial ultrasound and clinical assessment for remote and rural communities.** According to the current guidelines in the UK IV tPA can be given within 4.5 hours of stroke symptom onset. Mechanical thrombectomy can be performed within 6 hours of the onset of stroke symptoms; an extended time window of 6 to 24 hours from the time the patient was last known to be well can be offered in selected cases. Abbreviations: AF—atrial fibrillation; ECG—electrocardiography; ICH—intracranial haemorrhage; IV tPA—intravenous thrombolysis with tissue plasminogen activator; LMWH—low molecular weight heparin; LVO—large vessel occlusion; SBP—systolic blood pressure (mmHg).

## Strengths and weaknesses of the study

We are the first study, to our knowledge, to develop a triage model combining transcranial ultrasound and clinical assessment that could select patients with LVO for mechanical thrombectomy in the pre-hospital setting. This would be useful in remote and rural areas because it is not possible to transport every patient with suspected stroke to a thrombectomy centre.

Non-contrast transcranial ultrasound is less complex than contrast-enhanced ultrasound and more feasible in the pre-hospital environment, although it is not widely used in the UK. Additionally, TCCS is a more accurate technique for detecting major intracranial arteries as it is based on anatomical orientation around landmarks rather than indirect parameters (depth of insonation, flow direction and orientation of the transducer) used by transcranial Doppler ultrasound, which makes our ultrasound findings more reliable [33].

One of the main limitations of our study is the small sample size which likely increased the possibility of overfitting of the regression model, particularly for diagnosis of haemorrhagic stroke. Additionally, 53% (57/107) of participants were recruited within the first 24 hours, and 10% (11/107) patients had been given IV tPA prior to transcranial ultrasound which may have led to some potential changes in transcranial ultrasound findings, for example, due to the possibility of thrombus resolution [34].

CTA was not routinely performed in the initial study site—Raigmore Hospital, which largely serves a rural area—because mechanical thrombectomy was not widely available in Scotland at the time this exploratory work was performed. A further drawback is the lack of

diffusion-weighted magnetic resonance imaging (DWI MRI) to confirm the diagnosis, however DWI MRI is also unlikely to be available at short notice in other local hospitals in rural areas.

The two main limitations of transcranial ultrasonography are insufficient acoustic window [10] and user dependency, which prevent this imaging test from having wider clinical usage [35, 36]. We documented insufficient temporal acoustic window in about 17% of patients which is in agreement with previously published literature, and this is generally more common in elderly patients, which from a large proportion of stroke patients, females and an Asian population [35]. Further work to optimise ultrasound probe design, examining low-frequency insonation, an adaptive focusing method [37], and three-dimensional reconstruction of transcranial ultrasound images [38, 39] may be justified.

All patients in our study were able to tolerate the ultrasound scan; however, the median time to complete the transcranial ultrasound examination was 20 minutes, which is longer than previously reported [15, 16]. If transcranial ultrasound was performed before arriving at the hospital, this could result in treatment delays leading to poorer clinical outcomes unless the examination was performed while awaiting the arrival of patient transportation. Further protocol improvement could be helpful with examination of intracranial structures that can be easily performed.

The Regensburg project showed that major occlusion in the MCA can be detected with 90% sensitivity and 98% specificity with the use of contrast-enhanced TCCS and neurological examination in the hands of experienced stroke neurologists [40, 41]; the TCCS findings were compared with angiography which was not performed in our study. In contrast, we have demonstrated that non-contrast TCCS performed by non-experts can be feasible. The results of the Regensburg project are mostly applicable to urban settings for two main reasons. Firstly, experienced stroke neurologists are rarely available in remote and rural settings [44]. Secondly, we recruited a wide variety of patients with suspected stroke including acute ischaemic stroke as well as those who would not benefit from reperfusion therapy (TIA, stroke mimics and intracranial haemorrhages) which is more representative of the pre-hospital scenario.

## Implication of current findings for clinical practice

A simple and safe pre-hospital diagnostic test to enable selection of a proportion of patients who would benefit from rapid direct transfer for mechanical thrombectomy would improve access to treatment and reduce unnecessary transfers of acutely ill patients [42, 43]. A number of studies have been published that use mathematic modelling to estimate the likely benefits of different transport strategies for patients with suspected LVO—most commonly "drip-and-ship" and "mothership" models [44, 45]. We propose an approach where clinical assessment could guide the triage decision in case of insufficient acoustic window as we found that in about 17% of suspected stroke patients, transcranial ultrasound would have no diagnostic value (Fig 7). The advantage of the proposed model is that it allows assessment of potential eligibility for treatment and thus more beneficial transfer decisions.

For suspected stroke patients with possible LVO in a remote area with no CT or CTA available, the decision model may facilitate transfer directly to an endovascular centre, and avoid the unnecessary transfer of patients in whom tPA is contraindicated. Depending on transport times between the nearest tPA centre and endovascular centre, two triage transfer options may be considered [44]:

1. In a situation where the patient has absolute contraindications for IV tPA, or outwith the allowed time period for IV tPA, a direct transfer to the endovascular centre would be associated with a greater chance of a good outcome [44];

2. If the IV tPA and thrombectomy centres are far apart (at least 90 minutes transport time), a "drip-and-ship" model would be favourable because in this case patients would have a chance to benefit from IV tPA [44].

Pre-hospital tPA is currently used widely in rural areas for pre-hospital treatment of myocardial infarction [46] and could potentially reduce time to treatment but further data would be needed to ensure that absolute contraindications to tPA such as a subacute infarction, space occupying lesion, or intracranial haemorrhage can be safely excluded at the pre-hospital stage.

## Conclusion

We propose a triage model based on combined transcranial ultrasound and clinical assessment ("TUCA") for remote and rural communities to identify patients who could benefit from IV tPA and/or mechanical thrombectomy. Diagnostic accuracy of the proposed model for diagnosis of acute ischaemic stroke due to LVO was superior in comparison with clinical assessment alone, particularly in the first 24 hours post-stroke.

## Supporting information

**S1 Table. Sensitivity analyses of the regression models for likely diagnosis of intracranial haemorrhage and acute ischaemic stroke due to LVO based on transcranial ultrasound and clinical assessment.**
(DOCX)

## Acknowledgments

DA would like to thank the University of Aberdeen for providing an Elphinstone Scholarship to support her PhD project. The authors also thank Dr Neil Scott for giving advice on statistical analysis, Professor Jesse Dawson and Karen Shields for recruiting participants at the Queen Elizabeth University Hospital.

## Author Contributions

**Conceptualization:** Daria Antipova, Leila Eadie, Stephen Makin, Philip Wilson, Ashish Macaden.

**Data curation:** Daria Antipova.

**Formal analysis:** Daria Antipova.

**Investigation:** Daria Antipova.

**Methodology:** Daria Antipova, Leila Eadie, Stephen Makin, Philip Wilson, Ashish Macaden.

**Project administration:** Daria Antipova.

**Supervision:** Leila Eadie, Stephen Makin, Philip Wilson, Ashish Macaden.

**Validation:** Daria Antipova, Helen Shannon.

**Writing – original draft:** Daria Antipova.

**Writing – review & editing:** Daria Antipova, Leila Eadie, Stephen Makin, Helen Shannon, Philip Wilson, Ashish Macaden.

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
