## [Decision Letter · Decision Letter 0]

11 Sep 2020

The use of transcranial ultrasound and clinical assessment to diagnose ischaemic stroke due to large vessel occlusion in remote and rural areas.

PONE-D-20-18924

Dear Dr. Antipova,

We’re pleased to inform you that your manuscript has been judged scientifically suitable for publication and will be formally accepted for publication once it meets all outstanding technical requirements.

Kind regards,

Juan Manuel Marquez-Romero, M.D., M.Sc.

Academic Editor

PLOS ONE

1 . Please specify in the ethics statement in the Methods section and online submission information whether you obtained informed verbal or informed written consent from the relatives/legal representatives included in the study. If consent was verbal, please amend your current ethics statement to explain 1) why written consent was not obtained, 2) how you recorded/documented participant consent, 3) whether your ethics committee approved this consent procedure.

Reviewers' comments:

Reviewer's Responses to Questions

**Comments to the Author**

1. Is the manuscript technically sound, and do the data support the conclusions?

Reviewer #1: Yes

2. Has the statistical analysis been performed appropriately and rigorously? 

Reviewer #1: Yes

3. Have the authors made all data underlying the findings in their manuscript fully available?

Reviewer #1: Yes

4. Is the manuscript presented in an intelligible fashion and written in standard English?

Reviewer #1: Yes

5. Review Comments to the Author

Reviewer #1: The authors have clearly specified the investigation question. They have chosen the appropriate methodology to answer it. Enough safeguards have been taken to reduce possible bias. Statistical analysis is adequate. Data, tables and figures are clear and consistent. Conclusions are supported by the data reported and limitations of the study have been acknowledged.

6. PLOS authors have the option to publish the peer review history of their article (what does this mean?). If published, this will include your full peer review and any attached files.

Reviewer #1: No

---

## [Editor Report · Acceptance letter]

23 Sep 2020

PONE-D-20-18924 

The use of transcranial ultrasound and clinical assessment to diagnose ischaemic stroke due to large vessel occlusion in remote and rural areas. 

Dear Dr. Antipova:

I'm pleased to inform you that your manuscript has been deemed suitable for publication in PLOS ONE. Congratulations! Your manuscript is now with our production department. 

Kind regards, 

on behalf of

Dr. Juan Manuel Marquez-Romero 

Academic Editor

PLOS ONE